# Exploring the Spatial and Temporal Characteristics of China's Four Major Urban Agglomerations in the Luminous Remote Sensing Perspective

**Jiahan Wang [1], Jiaqi Chen [1,2,*], Xiangmei Liu [1], Wei Wang [3] and Shengnan Min [1]**

[1] College of Computer and Information Engineering, Hohai University, Nanjing 210098, China
[2] Aerospace Information Research Institute, Chinese Academy of Sciences, Beijing 100190, China
[3] Ping An Technology Company, Shanghai 200120, China
*   Correspondence: jiaqichen@hhu.edu.cn

**Abstract:** This study addresses the knowledge gap regarding the spatiotemporal evolution of Chinese urban agglomerations using long time series of luminescence remote sensing data. The evolution of urban agglomerations is of great significance for the future development and planning of cities. Nighttime light data provide a window for observing urban agglomerations' characteristics on a large spatial scale, but they are affected by temporal discontinuity. To solve this problem, this study proposes a ridge-sampling regression-based Hadamard matrix correction method and constructs consistent long-term nighttime light sequences for China's four major urban agglomerations from 1992 to 2018. Using the Getis-Ord Gi* hot-cold spot, standard deviation ellipse method, and Baidu search index, we comprehensively analyze the directional evolution of urban agglomerations and the correlations between cities. The results show that, after correction, the correlation coefficient between nighttime light intensity and gross domestic product increased from 0.30 to 0.43. Furthermore, this study identifies unique features of each urban agglomeration. The Yangtze River Delta urban agglomeration achieved balanced development by shifting from coastal to inland areas. The Guangdong-Hong Kong-Macao urban agglomeration developed earlier and grew more slowly in the north due to topographical barriers. The Beijing-Tianjin-Hebei urban agglomeration in the north has Beijing and Tianjin as its core, and the southeastern region has developed rapidly, showing an obvious imbalance in development. The Chengdu-Chongqing urban agglomeration in the inland area has Chengdu and Chongqing as its dual core, and its development has been significantly slower than that of the other three agglomerations due to the influence of topography, but it has great potential. Overall, this study provides a research framework for urban agglomerations based on four major urban agglomerations to explore their spatiotemporal characteristics and offers insights for government urban planning.

**Keywords:** remote sensing; long-term night-time light; spatiotemporal patterns; urban agglomerations

## 1. Introduction

Urban agglomerations (UAs) are highly developed forms of spatial organization, in which developmental connections exist between cities of different types and scales, and one or more metropolis(es) serve(s) as the local economic core [1]. The rise of UAs is an important sign of economic development, with a strong driving effect on economic development [2]. Over the past several decades, the urban proportion of the world's population has grown from 30 percent in 1950 to 55 percent in 2018 and is expected to reach 68% in 2050, with particularly large increases in Asia and Africa [3]. This process has caused a series of issues [4], such as ecological degradation [5], heat islands [6], and air pollution [7], particularly in developing countries.

China, as the world's largest developing country, experienced rapid economic growth after the 1992 Southern Tour, overtaking Japan as the second-largest economy [8], and it has

since undergone intensive urbanization since. To promote the country's economic development, the National Development and Reform Commission approved the development of seven UAs in 2018 [9], including the Yangtze River Delta Urban Agglomeration (YRDUA), Beijing–Tianjin–Hebei Urban Agglomeration (BTHUA), Greater Bay Area Urban Agglomeration (GBAUA), and Chengdu-Chongqing Urban Agglomeration (CCUA). These four major UAs encompass six provinces, four municipalities, and two special administrative regions, including Beijing, Shanghai, and Hong Kong, with a total area of 671,000 km$^2$ and a population of 420 million [10,11]. These values account for 30.0% of China's total population and 7.0% of the national land area, while the gross domestic product (GDP) of the region has reached 43.0 trillion yuan, thus accounting for 46.7% of the country's share and greatly contributing to China's economic development [12].

Research on the spatial and temporal expansion of UAs is beneficial for improving the accuracy of regional policies, constructing new development patterns, promoting the coordinated development of regional economies, and comprehensively promoting the further development of Chinese cities, the last of which is of great significance. Traditional methods generally use statistical data to quantitatively analyze UA expansion [13,14], but these data lack detailed spatial information. Remote sensing technology has been applied for UA detection due to its broad observation range, regular acquisition, and low cost [15,16], and Landsat data are widely used for urban extraction because of their high resolution and long-term continuity [17,18]. However, image preprocessing and classification extraction are time-consuming and labor-intensive, thus rendering them unsuitable for investigating the spatiotemporal evolution of large-scale UAs [19]. Luminous remote sensing, which detects night lights and thus reflects information about human activities, has been applied by many scholars to the study of spatial pattern changes in UAs [20,21]. Zhao et al. proposed a sigmoid function between the Defense Meteorological Satellite Program and processed Visible Infrared Imaging Radiometer Suite data to characterize their relationship [22]. Li et al. used the sigmoid function to transform VIIRS observations into data similar to DMSP, generating consistent global DMSP NTL time series data from 1992 to 2018 [23]. Their harmonization of DMSP and VIIRS nighttime light data from 1992 to 2021 at the global scale, which they published, has great potential application in monitoring urbanization changes and in studying light pollution and energy consumption [24].Due to the significant differences between the Defense Meteorological Satellite Program-Operational Linescan System (DMSP-OLS) data and the National Polar-orbiting Partnership Visible Infrared Imaging Radiometer Suite (NPP-VIIRS) data, the question of how to generate a set of night light remote sensing images over a longer time span is one of the key issues currently faced by researchers.

To solve this problem, we preprocessed DMSP-OLS data based on the Ridgeline Sampling Regression (RSR) algorithm and proposed a Hadamard matrix correction method, thus generating a luminous time series for 1992–2018. Population growth and digital elevation model (DEM) data were introduced to explore their connections with luminous growth. The hot–cold spot and standard deviation ellipsoid (SDE) methods were used to detect the built-up area expansion and directional evolution process of UAs, respectively. We also combined Internet big data and used the Baidu search index to study the connection between cities within UAs. This study explored the development trends of UAs by comparing the characteristics of the Four Major Urban Agglomerations (FMUAs), which could be meaningful for promoting rational planning of future UAs.

## 2. Materials and Methods

### 2.1. Study Area

This study focused on the FMUAs in China, which are relatively developed: the YRDUA, GBAUA, BTHUA, and CCUA, as shown in Figure 1. YRDUA and GBAUA are located on the southeast coast. The YRDUA is dominated by plains and hills, with a dense network of rivers, and it is located in the geographical center of East Asia, where it is the key to the East Asian route of the Western Pacific [25]. Backed by the interior and separated

by high terrain elevations in the north, the GBAUA is one of the most dynamic economic regions in the Asia-Pacific region [26]. The BTHUA is located in northern China and covers Beijing, the capital of China, and it is one of the largest and most developed UAs [27]. The CCUA is in southwest China, dominated by mountainous and basins, with a DEM greater than 1000 m in most areas, and it is the only inland urban cluster of the FMUAs [28].

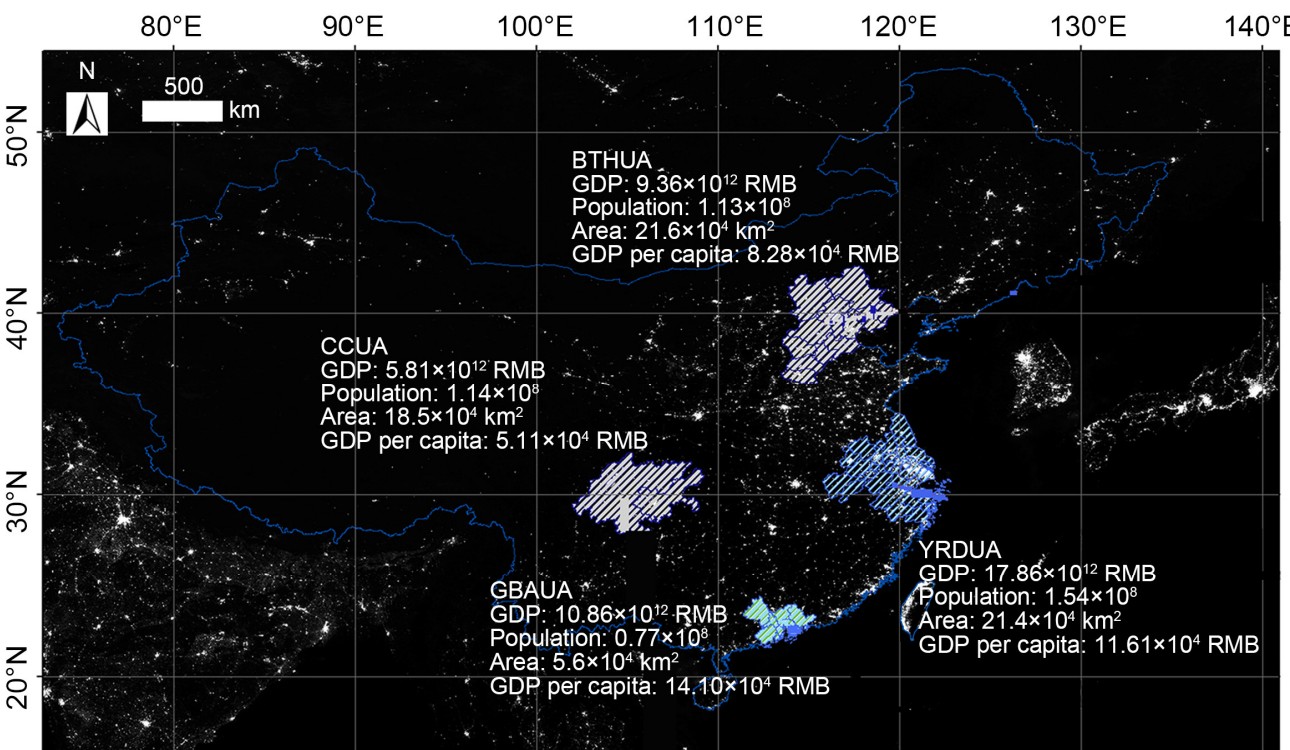

**Figure 1.** A night light view of the location and range of the study area. All population, GDP, area, and urbanization data in the figure are based on the 2018 statistics released by the local statistical bureaus.

## 2.2. Datasets

### 2.2.1. Night-Time Light

DMSP-OLS and NPP-VIIRS data were used as luminous remote sensing data in this study. Owing to its excellent photoelectric amplification, DMSP-OLS data can capture night-time near-infrared radiation from the earth's surface, including faint reflected light, and they were the only free and open-source type of night-time remote sensing data with worldwide coverage until 2012 [29,30]. After 2012, NPP-VIIRS inherited and enhanced the night light detection capability of DMSP-OLS. The spectral resolution of NPP-VIIRS is increased from 6 bits to 14 bits, and the spatial resolution is increased from 800 m to 500 m; therefore, this type of data gradually replaced the DMSP-OLS as new experimental data for researchers [31]. To maintain consistency between the two datasets, we selected annual raster image data from DMSP-OLS from 1992 to 2013, monthly NPP-VIIRS nocturnal data from 2012 to 2013, and monthly NPP-VIIRS nocturnal data from 2012 to 2013. The annual data of NPP-VIIRS were chosen for the luminous images after 2012.

### 2.2.2. Baidu Search Index

The Baidu search index reflects the search scale and frequency of a certain keyword regarding local Internet users (e.g., Beijing, Shanghai, etc.) within a given period of time [32]. The more frequently that one city is searched for as a keyword in another city, the more that this searching indicates the relevance of the former city, also reflecting the strength of the connection between the two. At the same time, it is assumed that the closer that the distance between the two cities is, the more convenient that the transportation is, the better that the economic mobility is, and the stronger that the connection between these two cities

will be [33]. We used high-performance web spider programs to automatically obtain this index and combine it with city distances to present correlations between cities.

*2.3. Methodology*

As shown in Figure 2, DMSP-OLS corrected with NPP-VIIRS was constructed as a consistent night light long time series from 1992 to 2018. Hot–cold spot analysis, SDE, and the Baidu search index were used to study the spatiotemporal evolution characteristics of FMUAs.

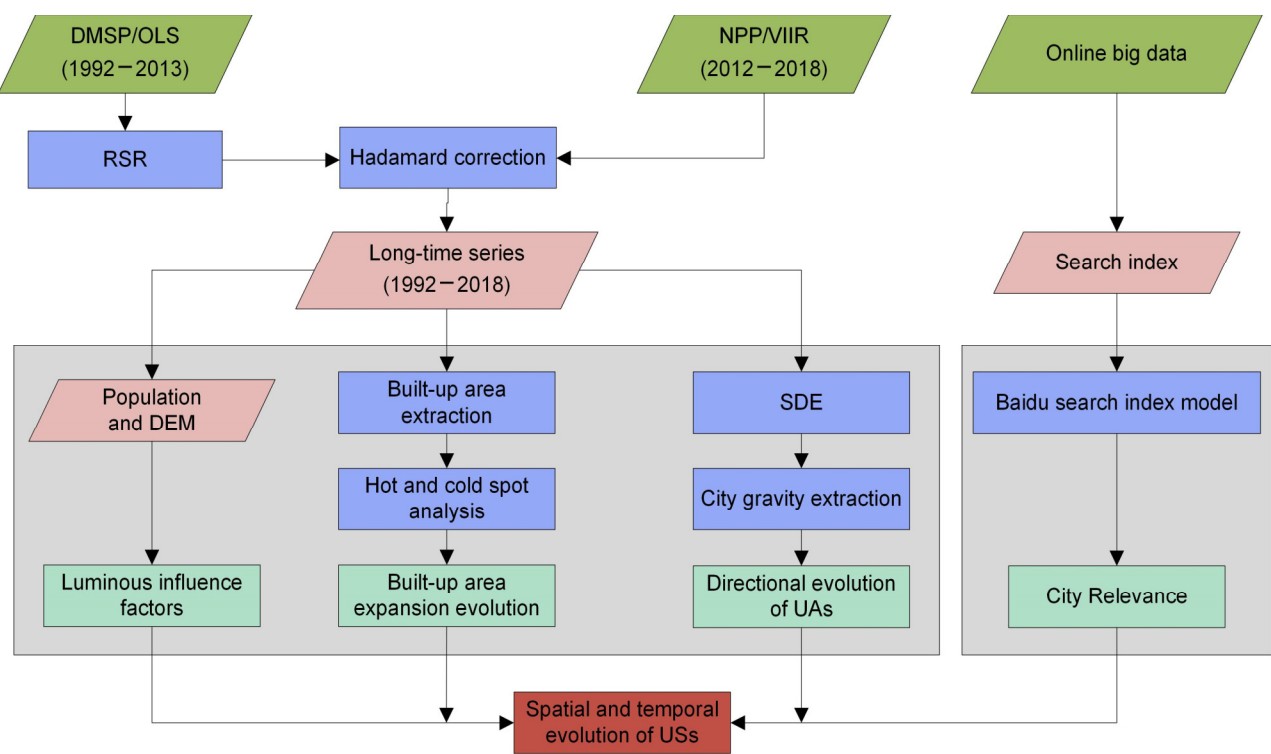

**Figure 2.** Flow chart of spatiotemporal characteristics analysis of the FMUAs based on luminous images and the Baidu search index.

2.3.1. Night Light Data Correction

- Ridgeline sampling regression

NPP-VIIRS has a higher spatial resolution than DMSP-OLS but is affected by problems, such as transient light source interference, negative values, and extreme values. Therefore, in this study, the annual NPP-VIIRS data were synthesized using the average value method and resampled to 0.5 km × 0.5 km for subsequent calibration. As a result, the resolution of the NPP-VIIRS data decreased and became the same as the DMSP-OLS data. At the same time, there were negative values affected by sensor sensitivity in the synthesized image; thus, the image had to be de-negatived. One resamples the original synthesized image at the grid level and sets the DN value of negative pixels to 0. For transient light sources, such as fishing boats and fire lights, which are still contained in the NPP image, the official released cloudless noise removal annual composite image data from 2015 and 2016 can be used to create masks to remove them.

In comparison to the NPP-VIIRS problem, DMSP-OLS is also affected by differences between different generations of sensors, resulting in inconsistent data. To study multitemporal satellite images, the differences between the satellites and the sensors must be minimized beforehand. There are many methods for correcting multi-temporal satellite images, such as histogram matching [34], pseudo-invariant feature recognition methods [35], and methods based on auxiliary data [36,37]. In multi-temporal image calibration, pseudo-invariant features must be identified, and then a calibration equation must be established

for the images to accurately represent the existing system bias. If performed manually, human errors or errors caused by different subjective interpretations can easily arise; therefore, automatic or semi-automatic methods, such as the ridge regression method, constitute a good solution [38]. RSR has the characteristics of high reliability in pseudo-invariant pixel recognition, high efficiency in mutual calibration, the potential for generalization of the experimental range beyond the calibration method used, and the minimization of bias in the calibration results [39].

To obtain a consistent long-term sequence image from DMSP-OLS night light images, we used RSR for calibration based on the average method and negative value removal. For different night light images, night light density maps are generated on the coordinate axis. The densest night light values remain relatively stable in the image, forming ridges, while the changes are concentrated beyond the ridges. The ridges reflect the relationships between different images, and a model is established for all the images.

To minimize the time interval and keep most of the pixels stable, we selected the F152000 data from the middle of the time series as the reference image and matched all the images to a uniform level. By utilizing the ridge relationship between the reference image and the target image density map and considering the systematic bias of night light values with uniformly distributed data points, a linear regression model was established:

$$DMSP_y = aDMSP_x + bDMSP_x{}^2 + c, \qquad (1)$$

where $DMSP_x$ and $DMSP_y$ represent the images before and after correction, respectively and $a$, $b$, and $c$ are the coefficients of least squares. Only 64 pairs of data points along the ridges were used to prevent overcorrection.

- Improved data correction combined with the Hadamard matrix

In imaging, there is non-uniformity between DMSP-OLS and NPP-VIIRS due to the influence of the sensor. Therefore, we propose a Hadamard matrix method to calibrate the consistency of these two sets of luminous data with different standards. First, the nocturnal values less than 0 in DMSP-OLS, which were processed by RSR, were excluded. Then, the 2012–2013 monthly night-time data of the two datasets were intercepted, and monthly ratios were calculated because of the temporal crossover of the data. The largest and smallest values of the 19 pairs of monthly ratio data were removed, and the final ratio image was obtained by averaging, with the results noted as FRI. The DMSP-OLS data were processed using the Hadamard matrix with the following equation:

$$CDS_i = FRI \odot RCDS_i, \qquad (2)$$

where $RCDS$ is the DMSP-OLS result processed by RSR, and $\odot$ is defined as:

$$A_{m*n} \odot B_{m*n} = \begin{bmatrix} a_{11}b_{11} & \cdots & a_{1n}b_{1n} \\ \vdots & \ddots & \vdots \\ a_{m1}b_{m1} & \cdots & a_{mn}b_{mn} \end{bmatrix}, \qquad (3)$$

### 2.3.2. Methods of Urban Spatial Expansion Analysis

- Hot and cold spot extraction for built-up areas based on Getis–Ord Gi*

Getis–Ord Gi* identifies high values (hot spots) and low values (cold spots) of spatial clusters and has been widely used to analyze biological habitats, epidemic diseases, areas of crime, etc. [40,41]. The specific calculation formula is as follows:

$$Gi_i * (Z) = \frac{\sum\limits_{j=1}^{n} w_{i,j}x_j - \overline{X}\sum\limits_{j=1}^{n} w_{i,j}}{S\sqrt{\dfrac{\left[ n\sum\limits_{j=1}^{n} w^2{}_{i,j} - (\sum\limits_{j=1}^{n} w_{i,j})^2 \right]}{n-1}}}, \qquad (4)$$

$$w_{i,j} = \begin{cases} 1, & distance \quad to \quad x_j < 100\text{km} \\ 0, & distance \quad to \quad x_j > 100\text{km} \end{cases}, \tag{5}$$

$$\begin{cases} \overline{X} = \sum_{j=1}^{n} x_j / n \\ S = \sqrt{\sum_{j=1}^{n} x_j^2 / n - (\overline{X})^2} \end{cases}, \tag{6}$$

where $Gi_j * (Z)$ is the Getis–Ord Gi* statistical z-score of the administrative city, which describes the spatial dependence of the city $j$ on the surrounding administrative city. $x_j$ represents the trend of the agglomeration area of city $j$, and $\overline{X}$ represents the average area of urban agglomeration. $w_{i,j}$ denotes the spatial weight of these two cities, and $n$ is the total number of pixels.

- Analysis of Built-up Area Expansion Index

To further investigate the expansion trend of built-up areas, expansion speed and expansion amplitude indices are introduced to effectively illustrate the changes in the cluster during the research time span. The specific calculation formula is as follows:

$$ES = \frac{ES_I - ES_i}{n}, \tag{7}$$

$$EI = \frac{EI_I - EI_i}{EI_i}, \tag{8}$$

where $ES$ represents the expansion speed of built-up areas, and $EI$ represents the expansion amplitude. $I$ and $i$ respectively represent the size of built-up areas in the $I$th and $i$th years, respectively.

- Directional evolution of UAs using SDE

To study the directionality of the UAs' spatial distribution, we introduced SDE, which is a classic method for analyzing the directional characteristics of spatial distributions [42]. Combining the night-time lighting coordinates, lighting scale, and development gravity of individual cities, we used SDE to study the spatial pattern of UAs in different periods, a method that can effectively illustrate the directional shift of urban spatial distribution. Based on SDE, it is possible to explore the centrality, extension, orientation, and spatial pattern of urbanization spatial distributions from a global spatial perspective. The formula is as follows:

$$SDE_x = \sqrt{\sum_{i=1}^{n} (I_{xi} - \overline{I_x}) / n}, \tag{9}$$

$$SDE_y = \sqrt{\sum_{i=1}^{n} (I_{yi} - \overline{I_y}) / n}, \tag{10}$$

where $I_{xi}$ and $I_{yi}$ are the pixel space center coordinates, and $\overline{I_x}$, $\overline{I_y}$ are the mean center coordinates. The azimuth angle is calculated as follows:

$$\tan\theta = (l_a + l_b) / l_c, \tag{11}$$

$$l_a = \left( \sum_{i=1}^{n} \overline{x}_i^2 - \sum_{i=1}^{n} \overline{y}_i^2 \right), \tag{12}$$

$$l_b = \sqrt{\left( \sum_{i=1}^{n} \overline{x}_i^2 - \sum_{i=1}^{n} \overline{y}_i^2 \right)^2 + 4\left( \sum_{i=1}^{n} \overline{x_i y_i} \right)^2}, \tag{13}$$

$$l_c = \sum_{i=1}^{n} \overline{x_i y_i}, \tag{14}$$

where $\overline{x_i}$ and $\overline{y_i}$ are the differences between the mean center and the coordinate axis, respectively. The equations for the long axis and short axis are as follows:

$$\sigma_x = \sqrt{2}\sqrt{\frac{\sum_{i=1}^{n}(\overline{x_i}\cos\theta - \overline{y_i}\sin\theta)^2}{n}}, \tag{15}$$

$$\sigma_y = \sqrt{2}\sqrt{\frac{\sum_{i=1}^{n}(\overline{x_i}\sin\theta + \overline{y_i}\cos\theta)^2}{n}}. \tag{16}$$

- Center of gravity index

The movement path of the development gravity center of a single city is defined by the movement of the luminous gravity center to highlight the urbanization development trajectory of each city in the UA [43], and the equation is defined as:

$$Lat_{kt} = \frac{\sum_{i=1}^{M_k} lat_i l_i}{\sum_{i=1}^{M_k} l_i}, Lon_{kt} = \frac{\sum_{i=1}^{M_k} lon_i l_i}{\sum_{i=1}^{M_k} l_i}, \tag{17}$$

where $Lat_{kt}$ and $Lon_{kt}$ are the longitude and latitude coordinates of the center of gravity of the night-time lights in region $k$ in year $t$, respectively. $l_i$ is the light value of the ith pixel. $M_k$ is the total number of pixels in region $k$. $lat_i$ and $lon_i$ are the latitude and longitude coordinates of the ith pixel point, respectively.

2.3.3. City Connection Based on the Baidu Search Index

We used big data to obtain the Baidu search index between two cities and then combined this index with the distance between them to calculate the city connection index. The model formula is as follows:

$$UER_{\Delta tij} = \frac{NBSI_{\Delta tij}NBSI_{\Delta tji}}{NDBC_{ij}NDBC_{ji}}, \tag{18}$$

where $UER$ is the urban economic relation. $\Delta tij$ represents the collection time interval for city $i$ and city $j$. $NBSI$ is the normalized Baidu search index, and $NDBC$ is the normalized distance between cities, which is defined as:

$$NBSI_{\Delta tij} = \frac{BSI_{\Delta tij}}{\max(BSI_{\Delta t})}, NDBC = \frac{DBC_{ij}}{\max(DBC)}. \tag{19}$$

where $BSI$ is Baidu search index, and $DBC$ is distance between cities. $\max(BSI_{\Delta t})$ represents the maximum search index value between a city and other cities, while $\max(DBC)$ represents the maximum distance value between a city and other cities.

## 3. Results

### 3.1. Results of Night Light Data Correction

As shown in Figure 3a, the DMSP-OLS data of FMUAs in 2011 were subjected to ridge regression and Hadamard matrix correction, respectively, resulting in Figure 3b,c. It can be seen from a comparison of the figures that the image noise and quality were poor before correction and were effectively resolved after correction.

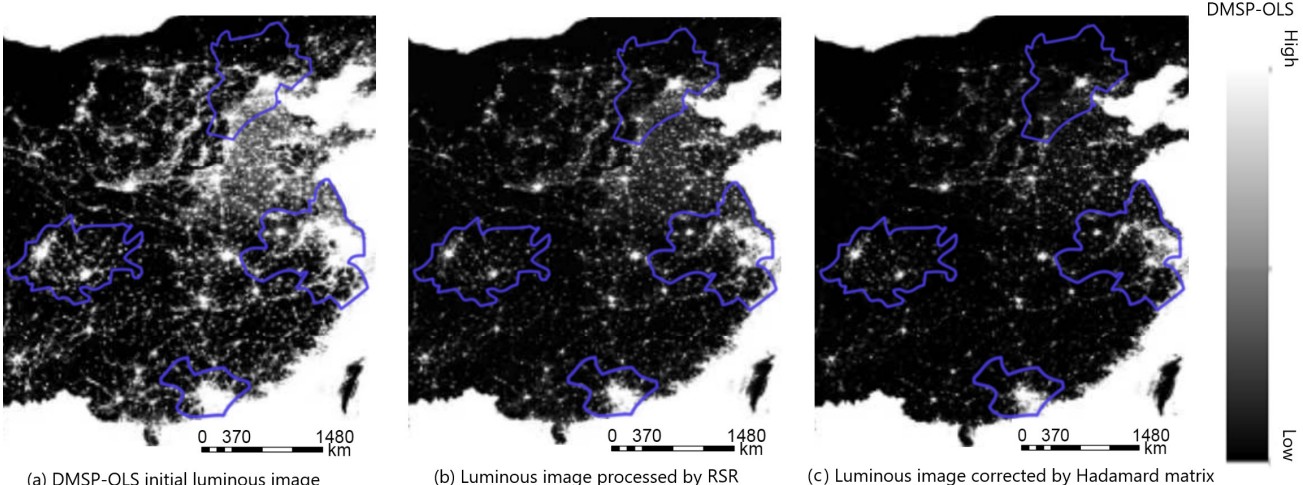

(a) DMSP-OLS initial luminous image   (b) Luminous image processed by RSR   (c) Luminous image corrected by Hadamard matrix

**Figure 3.** Results of DMSP-OLS correction processing. (**a–c**) The results of the images before DMSO-OLS processing, the images processed with RSR, and the results processed using Hadamard, respectively.

The DMSP-OLS data from 1992 to 2013 were corrected using RSR and constructed as a time series of total night-time light (TNL). Comparing Figure 4a with Figure 4b, it can be seen that the increase in TNL is smoother after the correction. TNL tends to have a strong correlation with economic activity; therefore, the GDP of the 52 cities in the study region was used to verify the accuracy of the corrected DMSP-OLS data by RSR. Compared with the original data, the corrected values have stronger correlations with the economy, with the correlation coefficient increasing from 0.3067 to 0.4309, indicating that the accuracy of the data improved after processing by RSR, as shown in Figure 4c,d. To verify the accuracy of the corrected data, this study conducted pixel-level validation of the DMSP-OLS corrected data, using Sentinel-2, a high-resolution satellite image with a resolution of 20 m, and manually extracted a built-up area as a pixel-level reference. The mean square error, peak signal-to-noise ratio, structural similarity index, and normalized cross-correlation were evaluated by comparing the two remote sensing images. The mean square error was used to compare the overall differences between the two images. The calculation method of the mean square error was to calculate the average of the square difference between pixels in the two images. The peak signal-to-noise ratio measures the ratio between the maximum possible power of the signal and the noise power that affects the fidelity of the signal. The structural similarity index evaluates the similarity of the structural information between two images. The normalized cross-correlation measures their similarity by calculating the cross-correlation between two images. The smaller that the mean square error is, and the higher that the peak signal-to-noise ratio is, the more similar that the images are. The range of the structural similarity index and normalized cross-correlation ranges from $-1$ to 1, with a value of 1 indicating complete similarity. The experimental results show that the mean square error is 6.2468, the peak signal-to-noise ratio is 40.1742, the structural similarity index is 0.7893, and the normalized cross-correlation is 0.7084. Based on these data, it can be concluded that the pixel-level accuracy of the corrected data is relatively high. On this basis, these DMSP-OLS corrected data were processed using the Hadamard matrix and constructed into a time series with the NPP-VIIRS data, as shown in Figure 4e. Greater connectivity between the two datasets was observed after processing, and there was no cliff drop, as with the original data. In particular, there was abnormally high TNL for NPP/VIIRS in 2014 and DMSP-OLS in 2010, a phenomenon not consistent with the actual TNL growth pattern that may have been generated due to unavoidable errors, such as haze and snow. However, these errors do not affect the overall correlation analysis. Thus, we conducted further studies on the UAs' evolution using the corrected images.

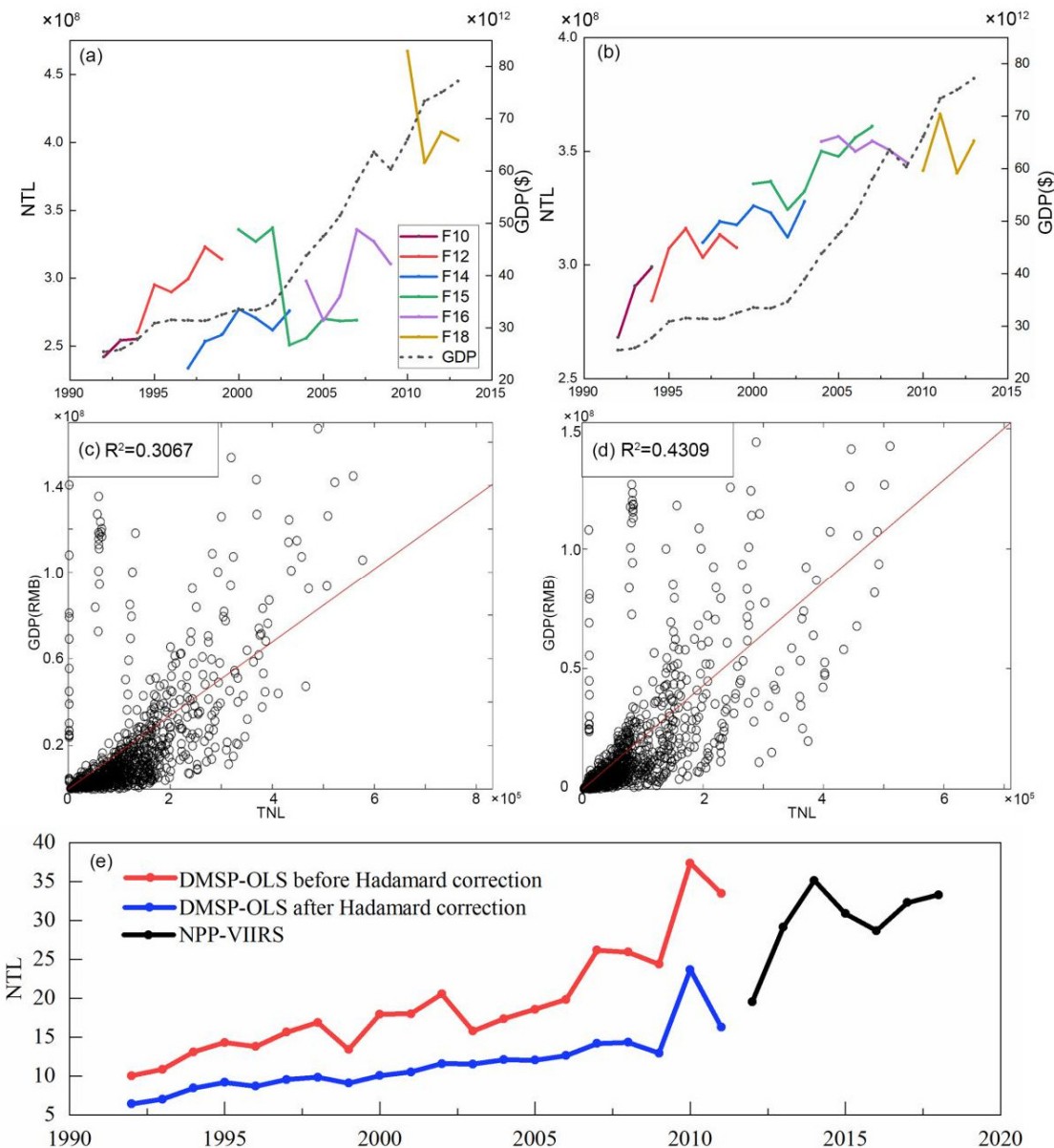

**Figure 4.** TNL time series results before and after correction processing. (**a**–**d**) Comparison of NTL series and GDP correlation coefficients before and after correction of DMSP-OLS data from 1992 to 2013 by RSR. (**e**) The 1992–2018 luminous long time series before and after Hadamard processing. The world GDP data was obtained from the World Bank.

### 3.2. Urban Evolution of FMUAs under Night Light

The above method was used to construct a long time series of FMUAs, and some of the image results are shown in Figure 5. The results show that the TNL of the FMUAs exhibited rapid growth. For a more precise analysis, the luminous mean and TNL were calculated for the YRDUA, BTHUA, GBAUA, and CCUA, respectively. Here, the TNL is totaled for all the luminous pixels in the study region and is used to describe the overall change in size of the UAs. The TNL is affected by the area and cannot be used to compare the development of each urban agglomeration; thus, it is necessary to average the TNL to obtain the mean value of night light.

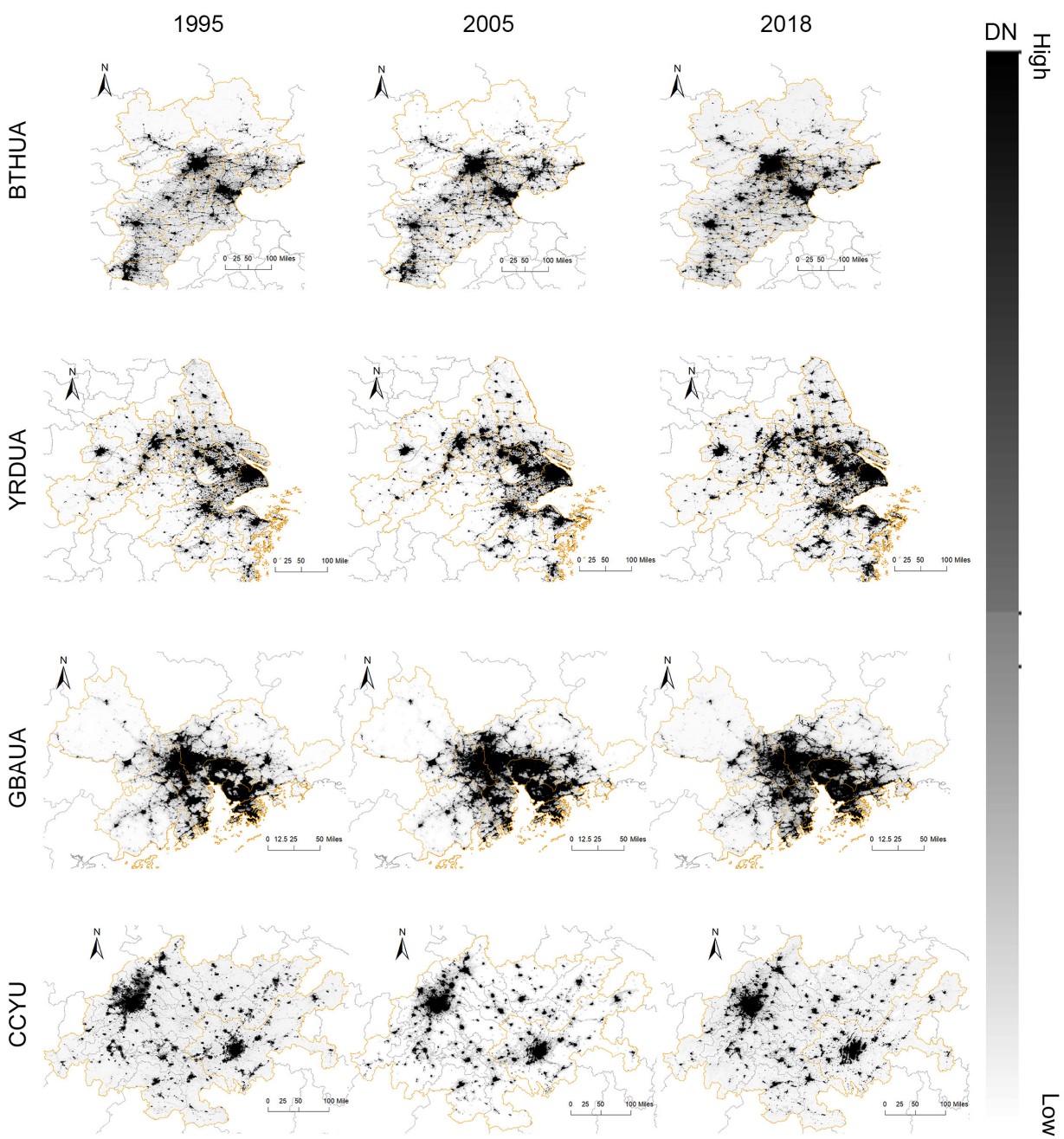

**Figure 5.** Night-time lighting images of FMUAs in 1995, 2015, and 2018.

The experimental results show that, from 1992 to 2018, the night lights of the FMUAs experienced rapid growth, and the UAs were further developed, as shown in Figure 6. The GBAUA has taken the lead in terms of its luminosity averages, but because of its smaller urban agglomeration area and earlier development, it has developed more slowly than the other UAs. In terms of TNL, the YRDUA has always been in the leading position, its luminous average being second only to the GBAUA. The BTHUA and CCUA are weaker than the former two UAs, especially the CCUA. The lighting distribution of these two UAs is also uneven. The BTHUA is concentrated in the southeast, especially Beijing and Tianjin, and the CCUA is mainly concentrated in Chengdu and Chongqing.

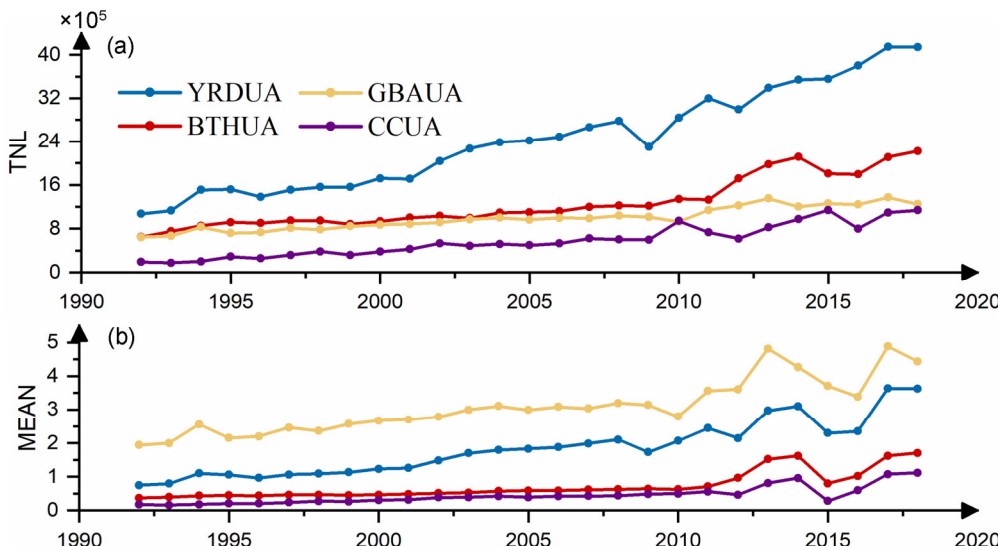

**Figure 6.** Results of total and mean values of night light for FMUAs, 1992–2018. (**a**) TNT of the FMUAs from 1992 to 2018. (**b**) Mean nighttime light value of the FMUAs from 1992 to 2018.

## 4. Discussion

### 4.1. Relationship between Population, DEM, and Luminous Growth

The growth of night light in UAs is often related to factors such as population flow and topography in cities [44].

The population data of each city were collected from 2002 to 2018, and the correlation coefficient for the night light reached 0.71, indicating a strong correlation between the population and night light. As shown in Figure 7, the population growth of the YRDUA is concentrated in the metropolises of Shanghai, Suzhou, Hangzhou, and Nanjing, while population outflow occurs in northern Jiangsu and southern Anhui. It is notable that there is significant population growth in Hefei in the west, which is also a rapidly growing region of night-time light in the western YRDUA. Population growth is most apparent in Guangzhou, Shenzhen, and Foshan in the GBAUA, with an annual net inflow of more than 400,000 people in recent years, while Hong Kong developed earlier, and its population growth is slow. The population increase in the BTHUA is mainly concentrated in Beijing and Tianjin, while the other cities are clearly growing slowly. Chengdu and Chongqing are the dual cores of the CCUA, with population growth concentrated in Chengdu. Correspondingly, Chongqing shows negative growth, in line with Chongqing's expansion from the core to the exterior.

Population and luminosity show some similarities, and there is also a strong connection between the development patterns of the UAs and their topography [45]. We extracted regional elevation maps of the FMUAs and applied elevation gradation to the DEM. Considering the DEM results together with the luminous growth model map, the results are shown in Figure 8 and Table 1.

In the YRDUA, GBAUA, and BTHUA, most of the night-time light is gathered in an area with a DEM of 0–200 m, and more than 90% of the plain area, with a DEM of 0–100 m, forms the night light growth region. The YRDUA is located on a low and flat terrain with a long coastline. Shanghai, Suzhou, and Nanjing, which are located on the plain, are developing rapidly. This plain extends all the way westward to Hefei, accelerating the development of inland areas such as Hefei and allowing for the balanced development of this urban agglomeration. However, Xuancheng, Jinhua, and Anqing are not coastal cities and are situated on higher terrain, limiting their development. The northwest and northeast parts of the GBAUA are isolated by high mountains and thus have insufficient space for outward expansion. Therefore, the entire GBAUA is naturally developing inward in a "fusion" pattern, and because of its small area, this growth is more concentrated in the center. The BTHUA is close to the Yanshan Mountains in the north, while the south is

flat, and its luminous growth is more evident. The night light is gathered in Beijing and Tianjin, while Zhangjiakou and Chengde have little night light growth due to the higher terrain, meaning that the UA's development is concentrated in the southeast. The CCUA is dominated by mountains and basins, making it significantly different from the other three UAs, and the terrain is largely above 100–1000 m. Chengdu is located in the Sichuan basin, known as the "Land of Heaven" [46], and Chongqing is situated in a mountainous area at the confluence of two rivers. The CCUA is concentrated in Chengdu and Chongqing, with Chengdu tending to contract inward, while Chongqing is expanding outward. Unlike Chengdu, Chongqing is not located in the basin. Its development is not as apparent as that of Chengdu, and the luminous growth of the flatter areas between these two places is also weaker.

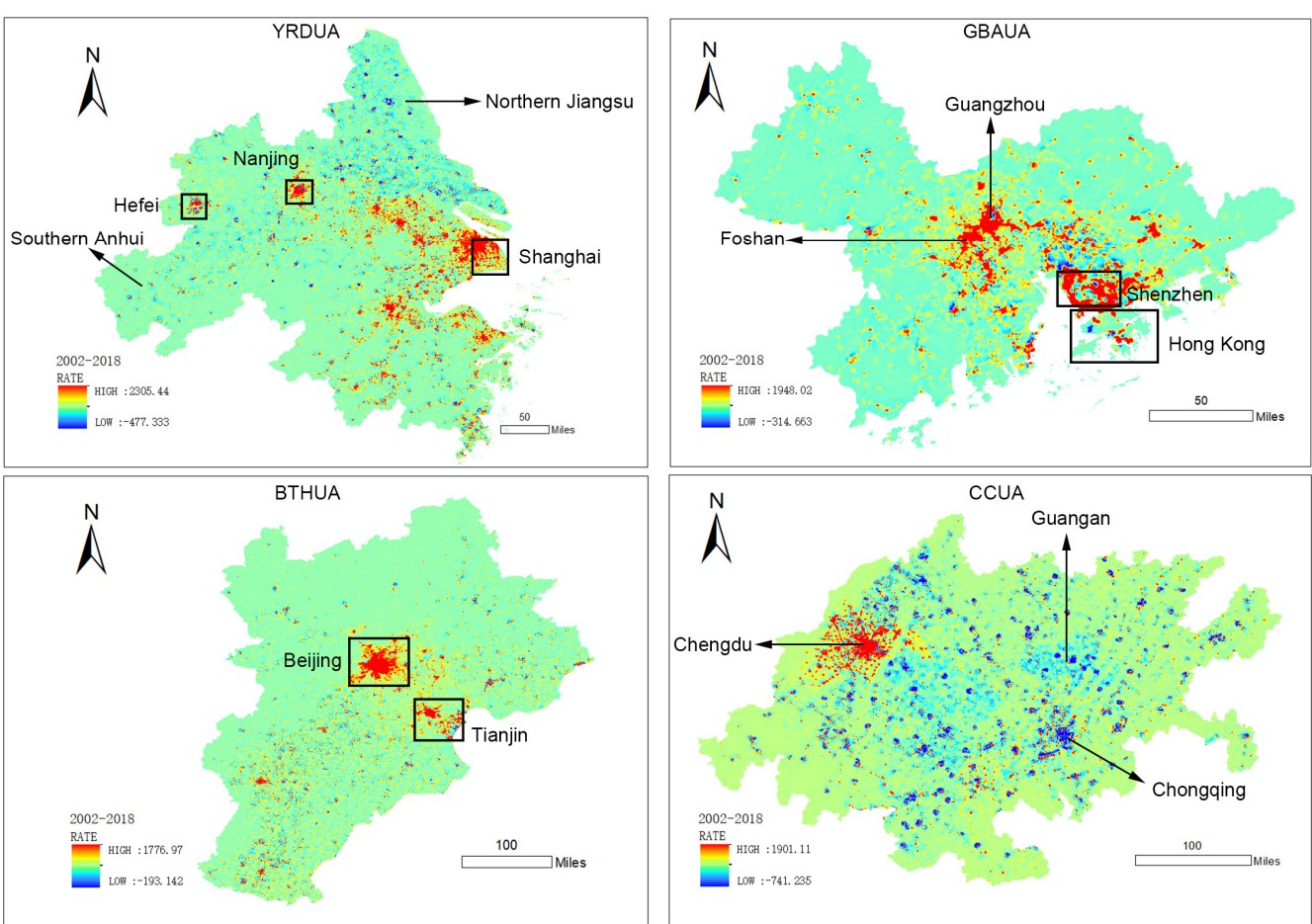

**Figure 7.** Population growth rates of FMUAs from 2002 to 2018.

**Table 1.** Table of luminous growth for terrain classification.

| A, B/% | YRDUA | GBAUA | BTHYA | CCUA |
|---|---|---|---|---|
| 0−100 | 73.10, 90.44 | 43.31, 90.44 | 61.07, 97.19 | 0.01, 0 |
| 100−500 | 21.37, 4.95 | 14.28, 4.95 | 34.47, 2.8 | 55.24, 76.25 |
| 500−1000 | 5.15, 3.77 | 19.34, 3.77 | 4.26, 0.02 | 29.47, 23.4 |
| 1000−1500 | 0.37, 0.76 | 18.29, 0.76 | 0.2, 0 | 8.3, 0.29 |
| 1500−3000 | 0.01, 0.09 | 4.79, 0.09 | 0, 0 | 6.99, 0.06 |

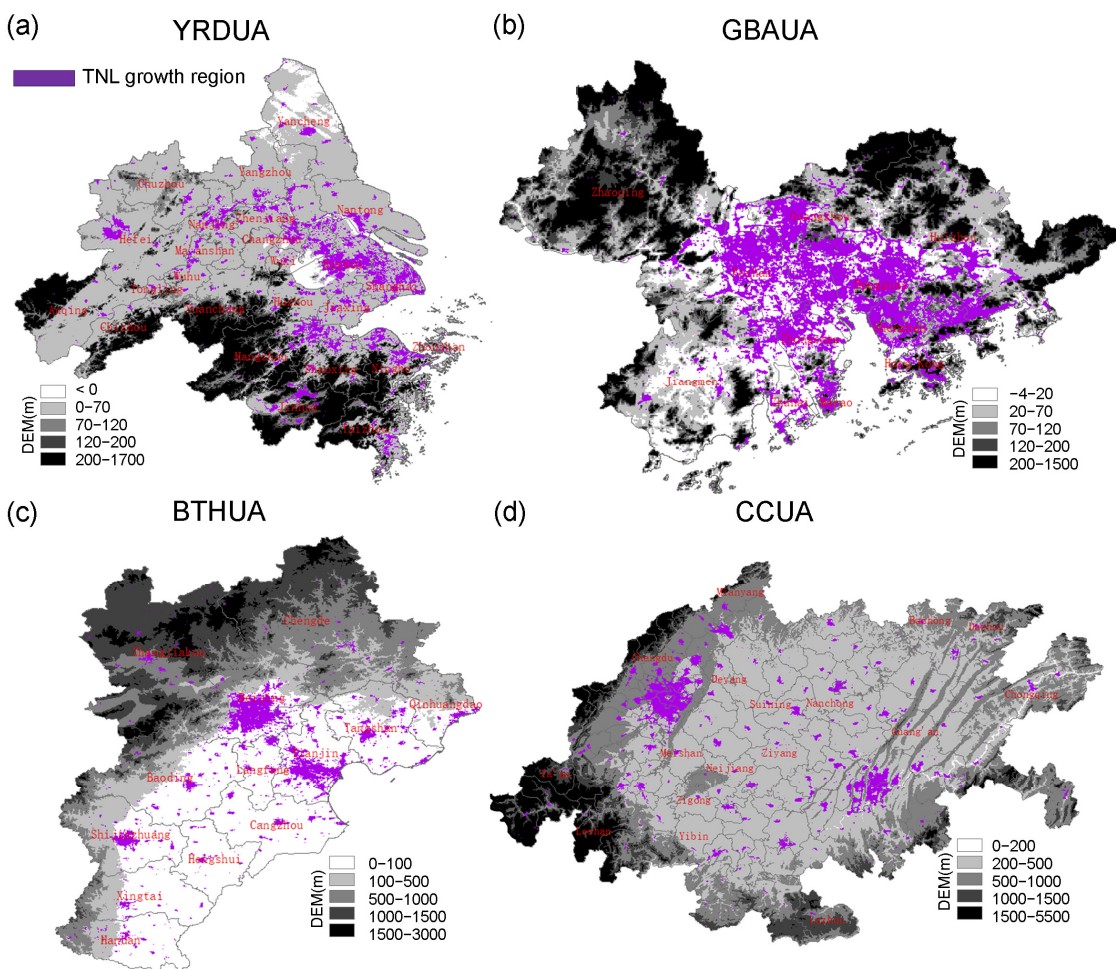

**Figure 8.** The results of the luminous growth model and DEM for the UAs. The purple part is the luminous growth region. (**a–d**) Display the results of the DEM and luminous growth model for YRDUA, GBAUA, BTHYA, and CCUA from 1992 to 2018.

### 4.2. Evolution of Built-Up Area Expansion in UAs

Night lights are rapidly increasing in number in the plain and population inflow areas of the FMUAs, reflecting the development of the UAs, especially changes in the built-up regions. A built-up region is an area that has been planned and developed by the government through infrastructure. It was discovered that it is not easy to identify a single optimal threshold for accurately extracting different cities simultaneously [47]. Thus, we use the dichotomous iterative method to set different thresholds for the nighttime light data of different UAs. For the same city in different years, the same threshold is used. The resulting cluster area is compared with the results of the China Urban Construction Statistical Yearbook until the error between the two is small. At this point, the range is considered the cluster area of the FMUAs.

As shown in Table 2, considering the 2010 data results as an example, different UAs use different thresholds to bring them closer to the aggregation range of statistical data. At this time, the error of the FMUAs is within 5%. As shown in Table 3, taking the BTHUA as an example, the annual verification results of the built areas from 1992 to 2018 are listed. The average annual error of the FMUAs from 2008 to 2018 was within 2%. The years 1992, 2000, 2005, 2010, and 2018 were used as dividing years to obtain the range of built-up areas for each year, and overlay analysis was performed to obtain the areas of the FMUAs presenting significant expansion, as shown in Figure 9. Here, the expansion range and speed were introduced to quantify the evolutionary characteristics. The expansion speed reflects the

growth rate of the built-up area of urban agglomerations in units of time, and the expansion amplitude reflects the growth amount of the built-up area of urban agglomerations over a period of time.

**Table 2.** Accuracy of the range of built-up areas in FMUAs in 2010.

|  | YRDUA | GBAUA | BTHYA | CCUA |
|---|---|---|---|---|
| Threshold | 19 | 12 | 12 | 11 |
| Statistical values | 7225.92 | 4536.63 | 3564.53 | 2367.67 |
| Extracted values | 7408.57 | 4626.38 | 3429.45 | 2271.75 |
| Error | −2.53% | −1.98% | 3.78% | 4.05% |

**Table 3.** Extracted agglomeration area accuracy for BTHUA from 1992 to 2018 (square kilometers).

| Year | Threshold | Statistical Values | Extracted Values | Error |
|---|---|---|---|---|
| 1992 | 12 | 1510.83 | 1427.10 | 5.54% |
| 1993 | 12 | 1607.75 | 1485.80 | 7.58% |
| 1994 | 12 | 1664.54 | 1719.28 | −3.29% |
| 1995 | 12 | 1728.21 | 1790.62 | −3.61% |
| 1996 | 12 | 1755.34 | 1802.60 | −2.69% |
| 1997 | 12 | 1786.24 | 1798.09 | −0.66% |
| 1998 | 12 | 1787.06 | 1918.88 | −7.38% |
| 1999 | 12 | 1908.07 | 2072.96 | −8.64% |
| 2000 | 12 | 2336.87 | 2532.73 | −8.38% |
| 2001 | 12 | 2393.91 | 2488.83 | −3.96% |
| 2002 | 12 | 2570.63 | 2695.59 | −4.86% |
| 2003 | 12 | 2838.57 | 2585.71 | 8.91% |
| 2004 | 12 | 2930.39 | 2927.91 | 0.08% |
| 2005 | 12 | 3046.38 | 3269.71 | −7.33% |
| 2006 | 12 | 3211.18 | 3275.26 | −2.00% |
| 2007 | 12 | 3334.85 | 3368.78 | −1.02% |
| 2008 | 12 | 3479.79 | 3596.60 | −3.36% |
| 2009 | 12 | 3520.08 | 3598.72 | −2.23% |
| 2010 | 12 | 3564.53 | 3429.45 | 3.79% |
| 2011 | 12 | 3626.90 | 3809.62 | −5.04% |
| 2012 | 12 | 3722.25 | 3904.53 | −4.90% |
| 2013 | 12 | 3840.71 | 3911.39 | −1.84% |
| 2014 | 12 | 4015.68 | 3925.37 | 2.25% |
| 2015 | 12 | 4230.44 | 4516.27 | −6.76% |
| 2016 | 12 | 4483.57 | 4522.99 | −0.88% |
| 2017 | 12 | 4607.23 | 4540.50 | 1.45% |
| 2018 | 12 | 4709.88 | 4752.82 | −0.91% |

In analyzing the changes in the urban built-up areas, the results of analysis with different time scales all have their unique reference values and need to be comprehensively analyzed in conjunction with actual situations. Analyzing changes in the built-up area using different intervals of years can capture some important short-term changes, such as sudden events or policy changes, but it requires careful manual adjustment to capture significant changes on the ground and may be subject to random events that can lead to some trends being misunderstood. Using equal time intervals to present the indicators will render the data more regular, easier to compare and analyze, and beneficial for observing the long-term trends of the built-up areas. At the same time, the selection of equal intervals of different time scales can also have a significant impact on the analysis results. For instance, a 3-year interval can reveal rapid changes in urban construction, while a 7-year interval can provide a rough understanding of changes in urban construction, which is useful for long-term planning and decision-making. This paper uses a 5-year equal interval to study the mid-term changes in the built-up areas, which may have a certain reference

value for timely adjusting of urban planning and controlling urban of expansion, as shown in Figure 10.

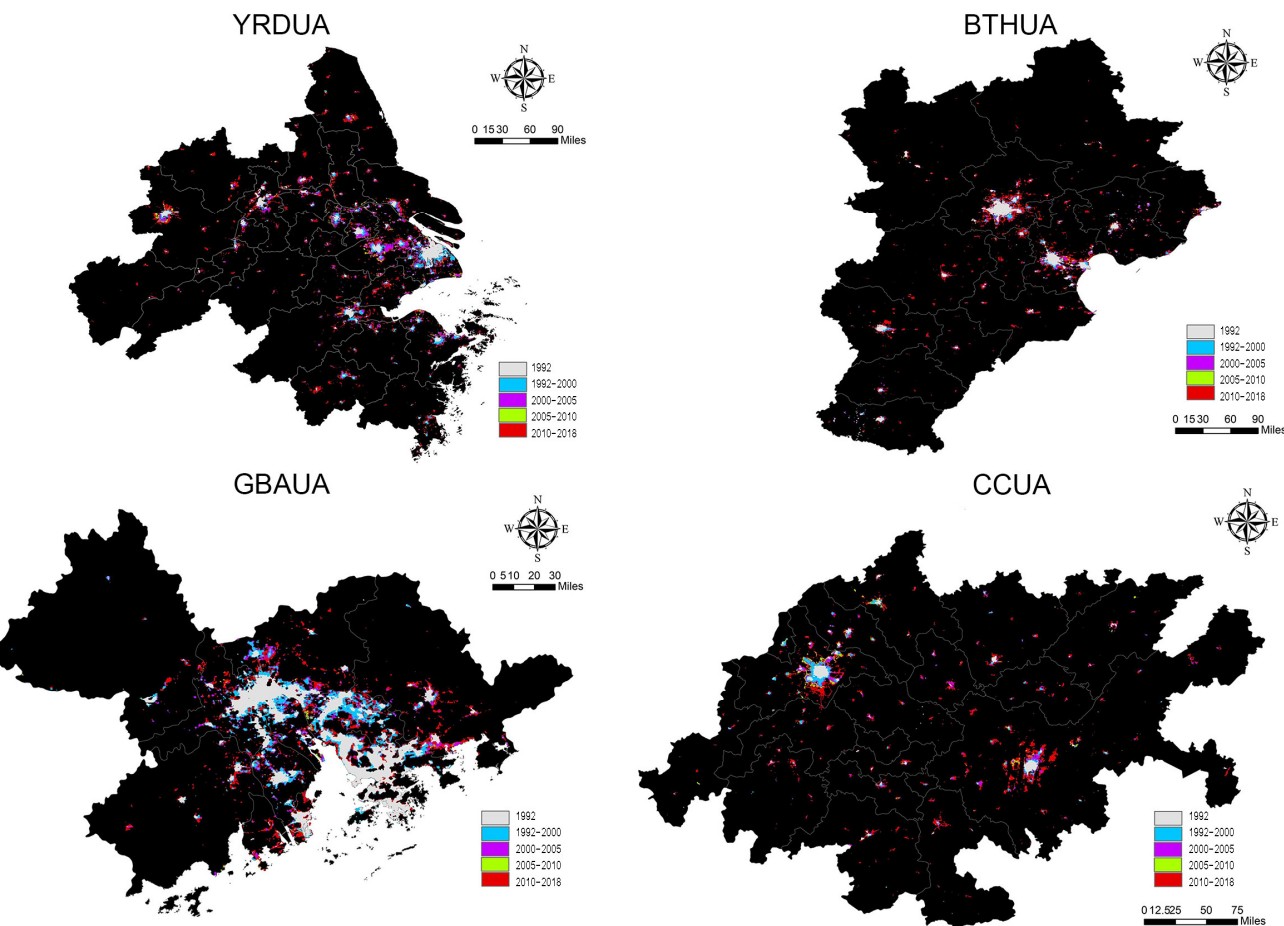

**Figure 9.** Expansion trends of FMUAs after threshold processing. By setting different thresholds, the built-up areas of FMUAs were extracted at different time periods.

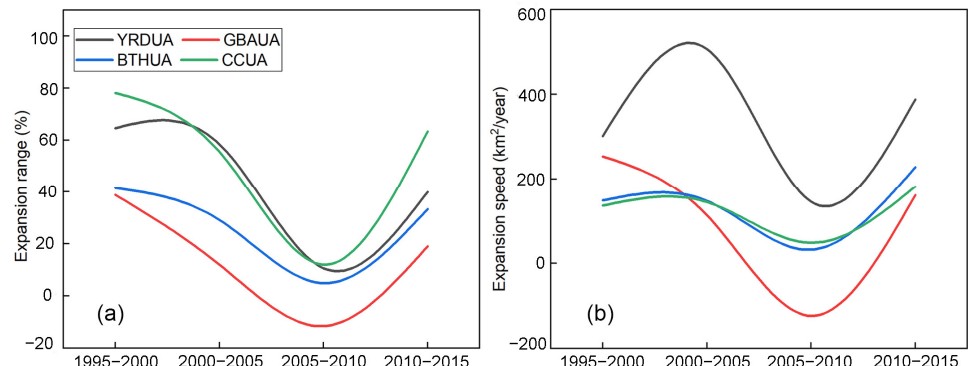

**Figure 10.** The expansion range and speed of FMUAs. (**a**) Analysis of the expansion range of FMUAs from 1995 to 2015. (**b**) Analysis of the expansion speed of FMUAs from 1995 to 2015.

Considering the expansion range, the FMUAs show a "U"-shaped pattern, which is consistent with urban development theory [48]. The expansion range of all four UAs declined in 2000–2010, while they reached extremely high levels in 1995–2000 and 2010–2015. The expansion speed reached its high points in 2000–2005 and 2010–2015. Similar to the expansion range, the expansion speed decreased in 2005–2010. During the previous expan-

sion period, the built-up area expanded faster than the rate of population urbanization, causing the construction of built-up areas to undergo a high-growth phase. The government's macro-control of land development led to a reduction in, or even contraction of, construction investment in built-up areas, which expanded at a high rate after it became more reasonable.

The expansion speed of the YRDUA was the fastest from 1995 to 2015, reaching an average of 335.4 km$^2$/year. However, in terms of the expansion range of the agglomerations, the CCUA had the most remarkable index, reaching about six times the range of the previous area after expansion by 2015. In 1995, Chengdu and Chongqing in the CCUA were still relatively backward in terms of development, with small built-up areas. When the speed of expansion slightly accelerated, there was a large increase in the expansion range of the CCUA. The overall development of the BTHUA and GBAUA was relatively slow. The GBAUA, which developed earlier, was affected by the terrain and the area, and its values were the lowest in terms of both the expansion speed and range.

Using Getis–Ord Gi* statistics, we selected a distance of 100 km and classified hot spots and cold spots above the 90% confidence level based on long-term trends in regional coverage. Dividing the time span into 1992–2005 and 2005–2018 can provide a long-term perspective, revealing changes in urbanization and land use patterns over a longer period, which is highly effective for studying urban cluster trends and identifying significant changes. Hot and cold spots were calculated based on the extracted built-up area ranges shown in Figure 11. The FMUAs were much more active in terms of expansion from 2005 to 2018 than from 1992 to 2005, indicating a time interval of high expansion.

The periods of 1992–2005 and 2005–2018 were analyzed as the first and second stages, respectively. In the first phase, YRDUA coastal areas, such as Shanghai, Suzhou, Ningbo, Wuxi, and Hangzhou, underwent rapid development, while the cities in Anhui located far from the coast, such as Chuzhou, Chizhou, Xuancheng, etc., developed at a slower speed. In the second stage, Shanghai and Suzhou in the coastal area still maintained high growth rates, and the development speed of inland cities in Anhui, such as Hefei, also increased rapidly. In both stages, the hot spots were concentrated in Shanghai and the surrounding regions and gradually transferred from the coast inland, thus achieving balanced development. The cold spots were concentrated in Xuancheng and Chizhou in southern Anhui, which could be related to the undulating terrain and population loss in these regions. Beijing and Tianjin were the main areas in both stages for the BTHUA, with Zhangjiakou and Chengde developing more slowly. Langfang, which is between Beijing and Tianjin, had clear regional advantages and became a hot spot at the same time as Beijing and Tianjin in the second phase. The development of the GBAUA in the first stage was concentrated in the middle areas, such as Dongguan, Guangzhou, Foshan, Shenzhen and Zhongshan, with hot spots in Guangzhou and cold spots in Zhuhai. In the second stage, the development expanded to the eastern regions such as Huizhou and Zhuhai, and the hot spots transferred to Huizhou. The development of built-up areas in Hong Kong and Macau was slow in both stages, which may be related to their smaller regional areas and mature development before 1992. The CCUA's development mainly focused on Guanan, Chengdu, and Chongqing, among which Guangan had a small volume in 1992. However, by 2000, the built-up area had expanded to reach four times the level observed in 1992. In the second stage, Chengdu and Chongqing developed rapidly, while Guangan stabilized. The hot spot areas were concentrated in Chongqing and its surrounding areas. However, Chengdu, which showed high expansion development, did not become a hot spot area in either phase, which may be related to its surrounding cities not undergoing high expansion development.

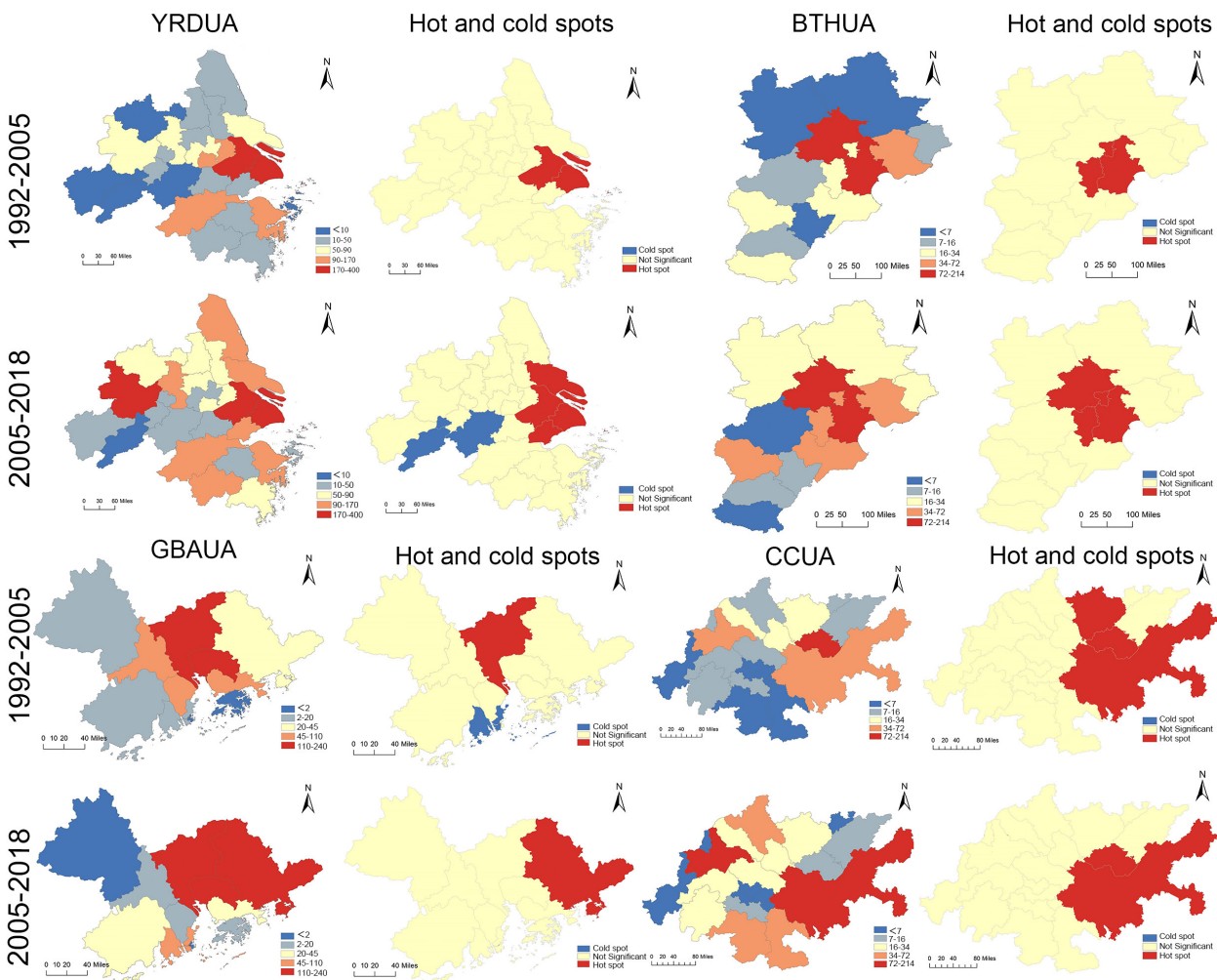

**Figure 11.** Distribution of hot and cold spots in FMUAs. The period 1992–2018 was divided into two time periods around 2005, where the diverging color scale in columns 1 and 3 represents the development speed of built-up areas, with blue to red indicating slower to faster development. The red areas in the second and fourth columns of the image are hot spots, while the blue areas are cold spots.

*4.3. SDE Directional Evolution of UAs*

One cannot explore the developmental characteristics of UAs without studying the directional trend of urban development. To explore the future development trends of the UAs, we analyzed the directional evolution characteristics of their urban spatial distribution using the SDE and gravity centers of individual cities. Using 1992, 2000, 2008, 2015, and 2018 as boundary years provided a more comprehensive understanding of the changes in the built-up area over time. As shown in Figure 12, the ellipses of the YRDUA and GBAUA show less significant changes, while the ellipses of Chengdu-Chongqing and the two major urban agglomerations of Guangdong, Hong Kong, and Macao show larger changes. The elliptical direction of the CCUA gradually transfers from east–west to north–south, and the ellipse undergoes a change, first shrinking and then expanding. The gravity center of the CCUA also shifts in the intermediate area of Chengdu-Chongqing. There is no significant movement in the gravity center of the GBAUA, but its elliptical spatial variation also shows a trend of first shrinking and then expanding.

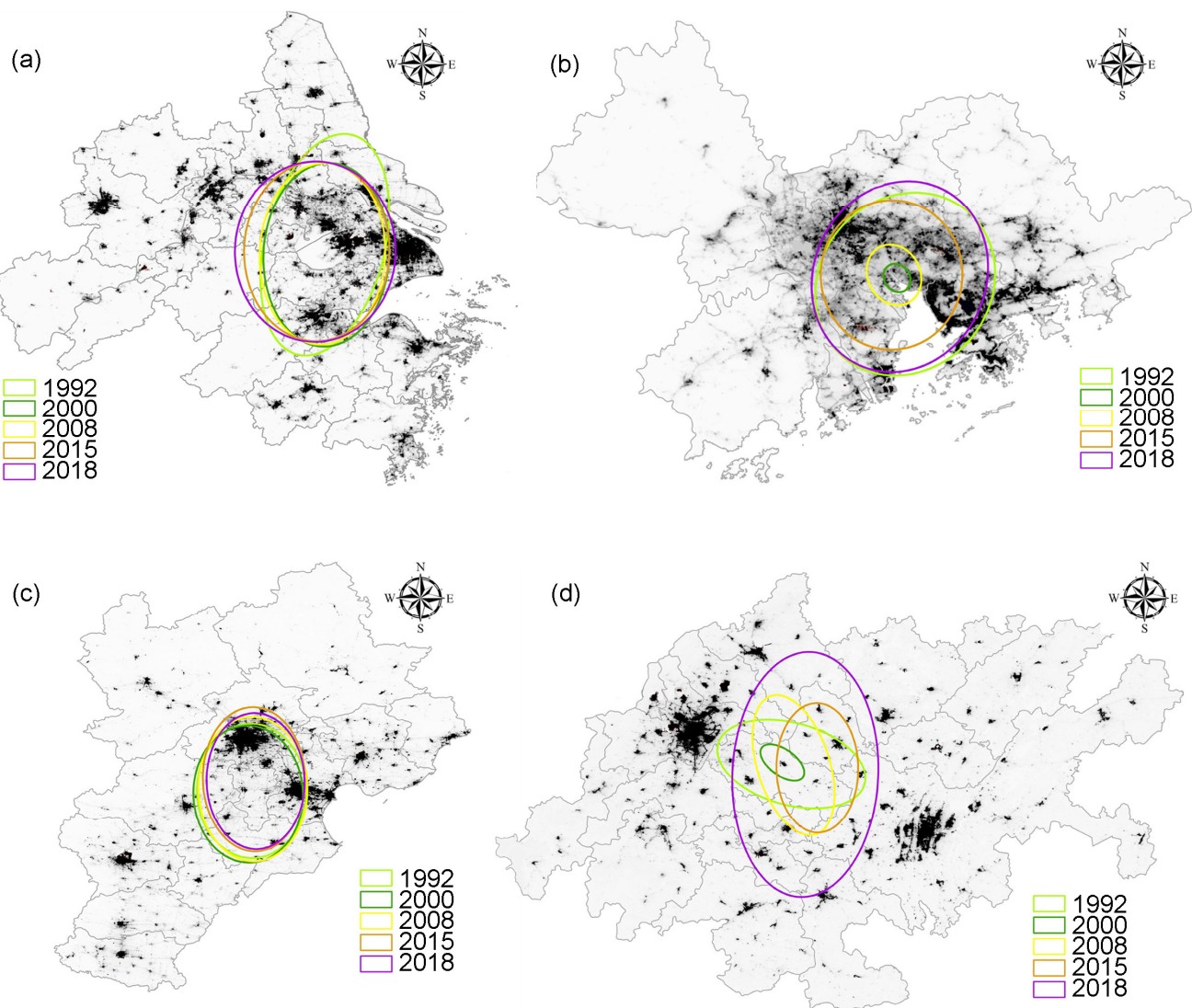

**Figure 12.** Standard deviation ellipsoid for FMUAs, 1992–2018. (**a**–**d**) The changing trend of the standard deviation ellipsoid for YRDUA, GBAUA, BTHYA, and CCUA from 1992 to 2018.

As shown in Table 4, the gravity center of the YRDUA is in the Taihu region, with the long axis increasing from 0.8224 to 1.0989 and the short axis decreasing from 1.5508 to 1.2357. This finding indicates that the YRDUA has shown a weak spreading trend in the east–west direction while shrinking in the north–south direction. This east–west expansion indicates a transfer of regional development from the coast to the interior, consistent with the results for the hot–cold spots. The azimuth of the GBAUA changed little from 1992 to 2008 but decreased sharply from 2008 to 2015, and the direction of development changed from northwest–southeast to northeast–southwest. The ellipse continued to expand after a rapid decrease in size in 2000, indicating that the surrounding small cities grew rapidly after 2000. The azimuth and long and short axes of the BTHUA did not change significantly, and its elliptical gravity center moved in small steps in the north–south direction while constantly changing among the three cities of Beijing–Tianjin–Baoding. The SDE area of the CCUA showed large fluctuations and decreased from 1992 to 2000, indicating a trend toward continuous concentration in its urban cluster development during this period. The ellipse area continued to increase from 2000 to 2018, and the concentration of the surrounding cities continued to weaken. The SDE results show a trend of clockwise rotation, and the development gradually changed from an east–west direction to a north–south spatial distribution pattern.

**Table 4.** Parameter characteristic values of SDE curves.

| UAs | Parameter | 1992 | 2000 | 2008 | 2015 | 2018 |
|---|---|---|---|---|---|---|
| YRDUA | Azimuth | −4.0268 | −19.0200 | −22.7213 | −40.8104 | −51.1588 |
| | Longitude | 120.3087 | 120.3296 | 120.2753 | 120.1984 | 120.1770 |
| | Latitude | 31.2796 | 31.1425 | 31.1539 | 31.1899 | 31.1845 |
| | Long axis | 0.8224 | 0.8602 | 0.8588 | 0.9942 | 1.0989 |
| | Short axis | 1.5508 | 1.2599 | 1.2434 | 1.2315 | 1.2357 |
| GBAUA | Azimuth | −1.1683 | 1.3552 | 1.9412 | −18.1176 | −2.1185 |
| | Longitude | 113.6484 | 113.5987 | 113.5802 | 113.5653 | 113.6160 |
| | Latitude | 22.7886 | 22.8238 | 22.8428 | 22.8435 | 22.8337 |
| | Long axis | 0.5241 | 0.0778 | 0.1610 | 0.4346 | 0.5276 |
| | Short axis | 0.5878 | 0.0913 | 0.1915 | 0.4591 | 0.5981 |
| BTHUA | Azimuth | 34.0014 | 25.9257 | 33.4617 | 37.1724 | 35.4692 |
| | Longitude | 116.4852 | 116.4326 | 116.5050 | 116.5219 | 116.5461 |
| | Latitude | 39.1014 | 39.0520 | 39.1301 | 39.2800 | 39.2506 |
| | Long axis | 0.8385 | 0.8402 | 0.8346 | 0.7848 | 0.7513 |
| | Short axis | 1.0332 | 1.0526 | 1.1097 | 1.1073 | 1.0455 |
| CCUA | Azimuth | 0.2717 | 0.7145 | 3.0877 | 72.3169 | −21.7125 |
| | Longitude | 105.0918 | 104.9962 | 105.1147 | 105.3669 | 105.2393 |
| | Latitude | 30.2869 | 30.3036 | 30.2734 | 30.2441 | 30.1756 |
| | Long axis | 0.4346 | 0.1380 | 0.3821 | 0.4329 | 0.7726 |
| | Short axis | 0.8043 | 0.2680 | 0.7715 | 0.6872 | 1.2974 |

As shown in Figure 13, to explore the development of cities within the FMUAs more precisely, we examined changes in the gravity center of each city to reveal the directional transformation of urban development. Analyzing the changes in the center of gravity of UAs every year is very useful for identifying areas undergoing rapid changes due to urbanization or other factors. The gravity center of the YRDUA shifted westward to the south from 1992 to 2002 and then northward until 2012, after which it finally shifted southward to the west. It can be noted that the gravity center gradually moved to the southwest and north because of the high degree of development of cities in the southeast of the Yangtze River Delta, such as Shanghai, followed by the rapid rise of Hangzhou, Nanjing, and Hefei. The gravity center of the GBAUA is located at the junction of Dongguan, Shenzhen, and Guangzhou. At the time in question, Hong Kong and Macau were more economically developed, but due to the rapid development of Guangzhou, the overall gravity center tended to develop in a northwest direction toward Guangzhou. It can be seen from the trend of the gravity center of each city that Foshan gradually shifted closer to Zhaoqing, and Shenzhen gradually shifted in the direction of Huizhou in the northeast, with general clustering in the middle. The gravity centers of Shijiazhuang, Xingtai, and Qinhuangdao in the BTHUA slowly moved towards Beijing, while Beijing and Tianjin gradually moved toward the northeast and were not close to the abovementioned areas. The BTHUA is still centered on Beijing, supplemented by Tianjin and Baoding, while Xiongan developed to take over the non-capital functions of Beijing. As the only inland city cluster among the FMUAs, the gravity center is located between Chengdu and Chongqing and has been hovering between the two for the last 27 years. These results show that the gravity centers of Chengdu and Chongqing have both moved in the direction of the northeast–southwest axis, while Mianyang and Leshan have gradually moved closer to Chengdu. Eventually, the CCUA will become a new, world-class city cluster with a unique dual-core development pattern.

The expansion of built-up areas and the evolution of the SDE both indicate that the FMUAs showed an expansion trend from 1992 to 2018, with the YRDUA, GBAUA, and BTHUA on the coast developing significantly more than the CCUA in the inland region. The YRDUA adopted Shanghai as its center and has gradually extended from the coast to Hefei inland, thus achieving a balanced development. The GBAUA has developed more slowly because of its earlier development. Although the BTHUA is developing rapidly, its development has been concentrated in the southeast coastal area, with Beijing and Tianjin

as the core and with serious regional imbalances. The CCUA is centered on Chengdu and Chongqing, which are in inland areas, and it is weaker in terms of development than the other three UAs.

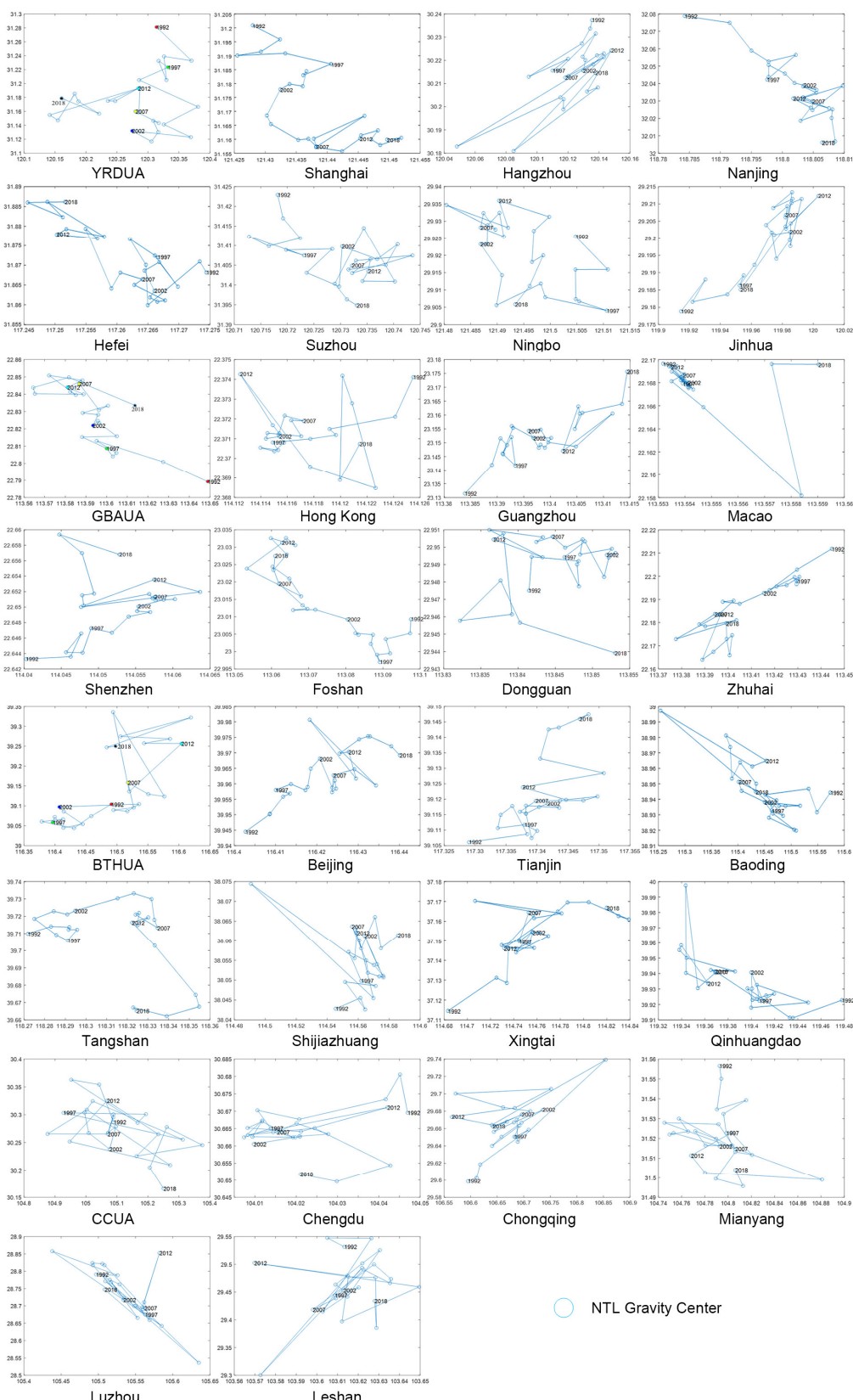

**Figure 13.** Changes in luminous gravity of selected cities in FMUAs, 1992–2018.

The characteristics of the FMUAs coincide with the development of China's economy today. The eastern region, especially the southeastern coastal region, has a developed economy and fast-growing urban agglomerations. While western China still has untapped development potential, the development of its UAs has been slow due to its geographical characteristics [49].

### 4.4. Development Planning of UAs Using the Luminous Correlation Model

The built-up area evolution and the SED results reveal the changing trends of the UAs, proving that night-time lights can better reflect the development of UAs as a whole. To specifically explore the connections between the respective cities, we used the Baidu search index to construct a correlation model of the FMUAs in 2018, as shown in Figure 14.

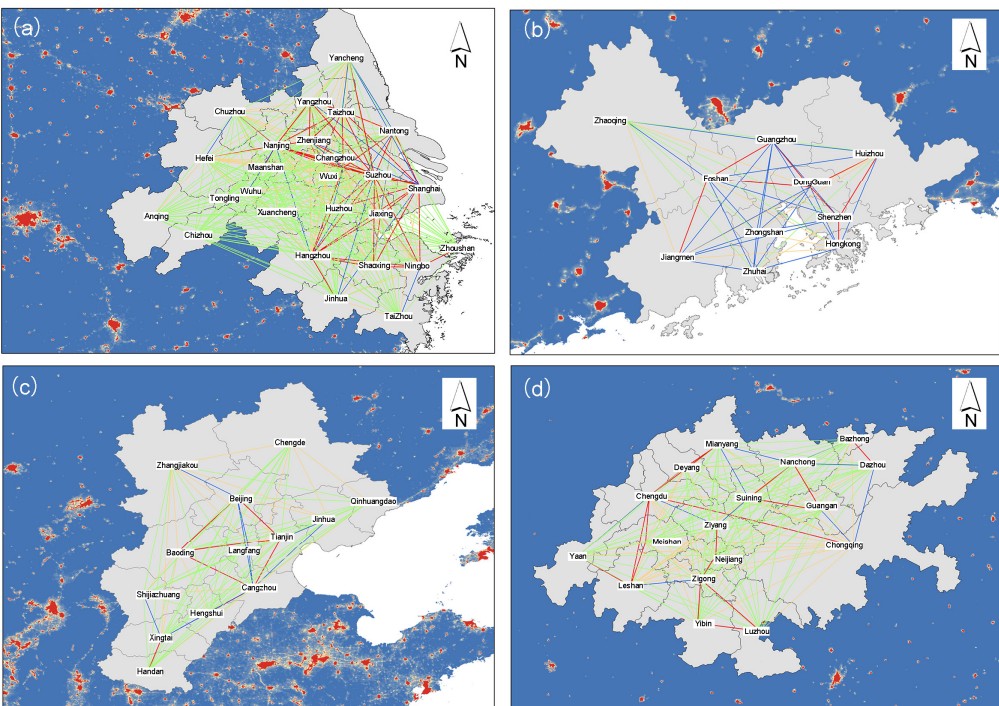

**Figure 14.** Visualization of the correlation degrees of the FMUAs. The ranking of the correlation degrees from weak to strong is shown with green, yellow, blue, and red lines. (**a**–**d**) The correlation models for YRDUA, GBAUA, BTHYA, and CCUA in the year 2018.

The total correlations of the YRDUA, GBAUA, BTHUA, and CCUA were calculated to be 391.46, 121.05, 67.48, and 40.65, respectively. The YRDUA has the highest degree of inter-city communication and the strongest overall degree of connection, while the CCUA is the weakest. In the YRDUA, Shanghai, Hangzhou, Suzhou, Nanjing, and Ningbo are the main cities with associations, of which Suzhou, as a high-GDP city, is also known as the "back garden of Shanghai", sharing a close connection with Shanghai. Nanjing, as the capital of Jiangsu Province, has close connections with Yangzhou, Zhenjiang, Maanshan, and Chuzhou. It can be noted that, as Anhui cities, Maanshan and Chuzhou are very close to Nanjing, which also shows that the development of the YRDUA is continuing to transfer inland. In general, several major metropolitan areas in the YRDUA are developing as a group. The GBAUA is dominated by Guangzhou–Foshan–Dongguan and Dongguan–Huizhou–Shenzhen, showing a double-triangle pattern. Shenzhen and Hong Kong, as two of the world's financial centers, have different political systems, but their close geographical locations mean that they have remained closely connected. Through "Hong Kong–Shenzhen cooperation" and the "Shenzhen–Shenzhen metropolis", these two connections could become deeper. In addition, Shenzhen also has a strong radiation effect on its surrounding cities and also has strong correlations with Huizhou and Dongguan.

Beijing, in the BTHUA, has a strong radiating effect on its surrounding cities and generates greater attraction from outside. Beijing–Tianjin–Baoding–Cangzhou show a square correlation trend, in which Langfang is located in the middle of the four cities and has close connections with all of them. Chengdu maintains a high degree of correlation with its surrounding cities in the CCUA. However, Chongqing, as a municipality directly under the control of the Central Government, is less connected to its neighboring cities, as it has a degree of independence.

The results regarding the changes in the UAs observed by night light and investigated using the city correlation model with the Baidu index are also consistent with the plans released by the government, as shown in Figure 15. The YRDUA forms a metropolitan area with Nanjing, Suzhou, Hangzhou, Ningbo, and Hefei as the major cities, and it radiates to the surrounding areas. Shanghai, as the key city, strengthens the economic link between Suzhou and Jiaxing and acts as the engine of the YRDUA to drive the development of the whole city cluster. The GBAUA takes Guangzhou, Shenzhen, Hong Kong, and Macau as its four core cities, constituting the main axis that supports interconnection within the city cluster. This situation will transform the "Guangzhou–Shenzhen–Hong Kong" tripod into an outward-oriented development model and create two mature metropolitan areas: "Guangzhou–Foshan" and "Shenzhen–Dongguan–Huizhou". This change will promote the reorganization of the city cluster pattern and create a "multi-core, strongly connected" world-class urban agglomeration. With Beijing as the core, the BTHUA will focus on strengthening connections with cities in the southeast. The CCUA will gradually move from a "double-core" model to a process of integrated development with the surrounding cities, centered on Chengdu and Chongqing. A trend will appear whereby Chengdu accelerates eastward, while Chongqing accelerates westward in its development. The CCUA will change from a "backward development" to "mutual development" pattern regarding the two cores and further capitalize on its dual-core driving ability to form two urban development economic zones. The future planning and development of the FMUAs are basically consistent with the experimental results, d the luminous remote sensing results have significance for the future planning and construction of UAs.

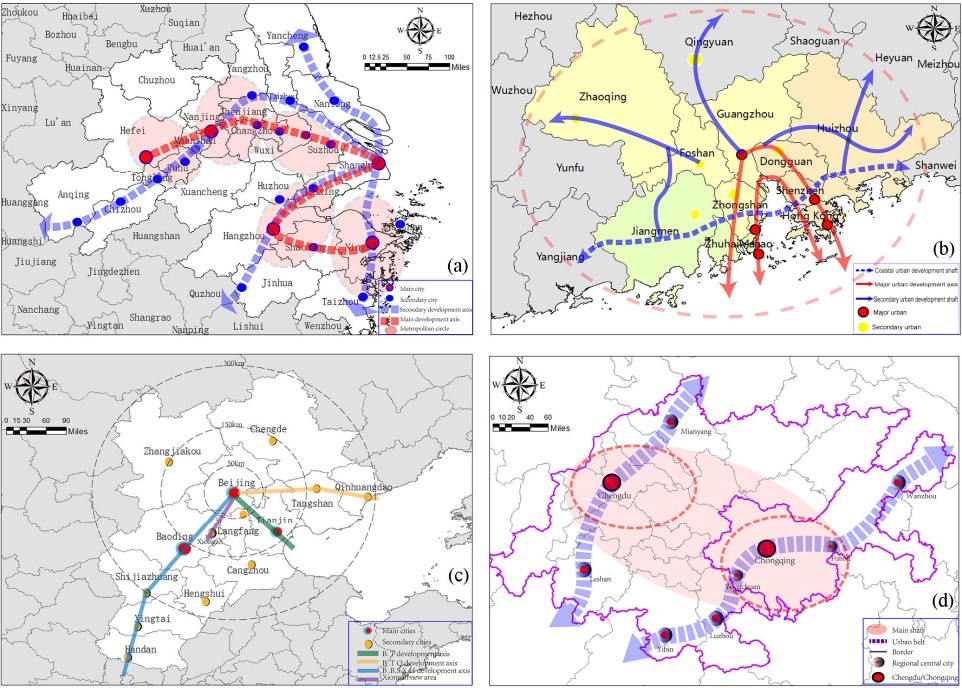

**Figure 15.** Government planning maps of FMUAs. (**a–d**) The government-released urban planning maps for YRDUA, GBAUA, BTHYA, and CCUA. The image source is from the development planning

documents released by the National Development and Reform Commission of the People's Republic of China and various provincial governments.

## 5. Conclusions

We constructed a long-term luminosity series spanning 1992 to 2018 for the FMUAs of the YRDUA, GBAUA, BTHUA, and CCUA. We found that population and topography have important influences on the expansion of UAs. The population is highly correlated with nocturnal growth, with a correlation coefficient as high as 0.71. Changes in topographic elevation limit the speed and direction of urban development and have especially decelerated the development of the CCUA, which is situated at a higher elevation.

When analyzing changes in urban built-up areas, different time scales have their unique reference values. A uniform time interval sequence will make the data more regular and easier to compare and analyze, which is conducive to observing the long-term trends of built-up areas but may not capture major changes that occur outside the set time interval. Non-uniform time interval sequences can better understand the phased development of UAs, but sudden events or policy changes require manual adjustments to capture significant changes on the ground, which is subjective. At the same time, the results obtained using different data points are inconsistent, making them difficult to compare and analyze. In this study, to ensure the robustness of the analysis, we comprehensively considered the advantages and disadvantages of different time scales for analysis and constructed a research framework for the evolution of built-up areas, providing valuable insights for the growth and development of UAs. The range of built-up areas was extracted from luminous images, and the expansion range and speed were used to measure the development of the FMUAs. The results show that the FMUAs conform to the "U" pattern, with the YRDUA and GBAUA on the southeast coast showing significantly greater development than the BTHUA and CCUA. Hot–cold spots were constructed based on Getis–Ord Gi* statistics, and directional evolution was determined using SDE. To study the inter-city correlations, a luminosity correlation model was constructed based on the Baidu search index. The study results indicate that the YRDUA, with Shanghai as the center, is continuing to develop inland and strengthen its associations with the inland cities, such as Hefei. The GBAUA developed earlier and shows less variability between cities but will continue to develop inwardly due to topographical constraints in the north and its small area. The BTHUA has always centered on Beijing and Tianjin, with uneven development. The CCUA, located in the western part of the interior, is developing more slowly. The cities around Chengdu continue to move closer to the center, while Chongqing continues to expand outward. With these two cities as the core, the rest of the regional cities form small metropolitan areas, developing together. These experimental results are consistent with governmental planning, and this research has important reference value for future urban cluster planning.

In summary, this study constructed consistent, long-term nighttime light remote sensing data and analyzed the directional evolution of urban agglomerations from multiple dimensions. The experimental results are consistent with government planning, and this research provides valuable references for the future planning of urban agglomerations. The directional evolution framework of urban agglomerations proposed in this study can be applied to other regions to obtain a better understanding of urban development dynamics.

**Author Contributions:** Conceptualization, J.W. and X.L.; methodology, J.C., X.L. and J.W.; software, X.L., W.W. and J.W.; validation, X.L., J.W. and S.M.; formal analysis, X.L., J.C. and J.W.; investigation, W.W. and S.M.; resources, J.C., J.W and S.M.; data curation, X.L., J.W. and W.W.; writing—original draft preparation, J.W., X.L. and W.W.; writing—review and editing, J.C., X.L. and J.W.; visualization, S.M., J.W. and X.L.; supervision, J.C.; project administration, J.C. and S.M; funding acquisition, J.C. All authors have read and agreed to the published version of the manuscript.

**Funding:** This work was supported by the National Natural Science Foundation of China (Grant NO. 61771183).

**Data Availability Statement:** The datasets generated during and/or analyzed during the current study are available from the corresponding author on reasonable request.

**Acknowledgments:** We are very grateful to the editors and reviewers who significantly contributed to the improvement of this paper. We would like to acknowledge the assistance of Yi Zhou in data processing.

**Conflicts of Interest:** The authors declare no conflict of interest.

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
