# Peer review of "Exploring the Spatial and Temporal Characteristics of China’s Four Major Urban Agglomerations in the Luminous Remote Sensing Perspective"

_remotesensing, doi:10.3390/rs15102546_

Round 1
Reviewer 1 Report
Comments:
1. I do not see the research gap in the abstract and introduction section.
2. Most of the results can be found in other studies, even though the authors adopted a new or different methods. I am wondering what the study contributes to the future research or development of UA.
a) Yang C, Xia R, Li Q, et al. Comparing hillside urbanizations of Beijing-Tianjin-Hebei, Yangtze River Delta and Guangdong–Hong Kong–Macau greater Bay area urban agglomerations in China[J]. International Journal of Applied Earth Observation and Geoinformation, 2021, 102: 102460.
b) Wang, C., Yu, B., Chen, Z., Liu, Y., Song, W., Li, X., Yang, C., Small, C., Shu, S., & Wu, J. Evolution of Urban Spatial Clusters in China: A Graph-Based Method Using Nighttime Light Data[J]. Annals of the American Association of Geographers, 2022, 112, 56-77
c) Zhao, X., Li, X., Zhou, Y., & Li, D. Analyzing Urban Spatial Connectivity Using Night Light Observations: A Case Study of Three Representative Urban Agglomerations in China[J]. IEEE Journal of Selected Topics in Applied Earth Observations and Remote Sensing, 2020, 13, 1097-1108
3. What does the “FMUA” mean?
4. It is difficult to read the Figure 4. First, what does the dashed line mean in (a) and (b)? Second, a city-level validation (c and d) is not enough, since you will used the corrected nighttime light data to extract built-up area at pixel level. I suggest that a pixel-level validation is required. Third, in (e), the DMSP-OLS before Hadamard correction is much closer to NPP-VIIRS, than the DMSP-OLS after Hadamard correction. Based on this worse result, why did the authors still adopted the Hadamard correction?
5. Since the nighttime light data were corrected to improve its consistency of time-series, why the authors still used different threshold value to extract different year’s urban built-up area. A full validation results of all years should be added in Table 3.
Reviewer 2 Report
The paper presents practice of data analysis with nighttime light and a search index for measuring urban growth in several dimensions. I understand the author spend much effort for variety of the outputs. However, I think the paper has not yet reach a level to publish in this journal. I would like the authors addressing issues below for next submissions.
1. Problems shall be specified, otherwise readers cannot understand innovative points.
2. I could not find any innovative or very new in the conclusions.
3. DBC and BSI in Equation 17 are not defined.
4. I found many typos, such as "VIIR", "SAR", “FMAUs".
5. "Expansion range" is not defined.
6. Figure 15 should indicate citation.
7. Figure 10: It is not fair to compare growth between periods of five years and eight years.
Reviewer 3 Report
This study proposed a Hadamard matrix correction method based on ridgeline sampling regression (RSR) to construct long-term nocturnal series of four major urban agglomerations (UAs) in 1992-2018, and then detected the evolution of four UAs via the Getis-Ord Gi* hot-cold spot extraction and standard deviation ellipse. And it further combined the Baidu search index to explore the city-relatedness within UAs. In general, the idea is original and the paper is well-written. However, some issues remain, I would recommend the authors revise the manuscript, with a keen eye to the issues raised.
1. In Section 3.1, it is recommended to compare DMSP-OLS data corrected by RSR with data corrected by other methods such as pseudo-invariant region method.
2. In Section 4.2, the author used the threshold method to extract urban areas in 1992-2018, while the spatial resolution of DMSP-OLS and NPP-VIIRS was different and the author mentioned “Therefore, the annual NPP/VIIR data is synthesized by the average value method and resampled to 0.5km×0.5km for subsequent calibration” in Section 2.3.1. The same issue is in Figure 8.
3. The data set from which the population gridded data comes needs to be indicated in Section 2.1.
4. The standardization of English abbreviations, such as DMSP, VIIRS, and FMUA.
5. Table 1 can be deleted.
6. “km2” should be “km2” in lines 48 and 370.
7. In Section 2.3.3, what are “DBC” and “BCI”?
8. In Figure 3, What is SAR?
9. In Figure 4(a)-(b), what is the meaning of the black dotted line.
10. In Figure 11, the meaning of the legend in columns 1 and 3 need to be shown.
11. The picture definition is low, thus, the administrative boundary of the city is not clear in the Figures.
12. The reference citation format needs to be standardized.
Round 2
Reviewer 2 Report
Thank you for the revisions.
2. We propose a new method of Hadamard matrix correction based on the ridgeline sampling regression algorithm to address the issue of discontinuity in long-term data.
If you highlight the method as an innovative point, I think you should refer the paper and dataset below to discuss advantages of your method.
A harmonized global nighttime light dataset 1992–2018 | Scientific Data (nature.com)
Harmonization of DMSP and VIIRS nighttime light data from 1992-2021 at the global scale (figshare.com)
4. "FMAUs" has been spelled out as "Four Major Urban Agglomerations" (Line 88).
Then "FMAUs" should be "FMUAs" like the other FMUAs.
5. "The expansion speed was obtained by dividing the different built-up ranges for different years by the intervals of time, while the expansion amplitude was obtained by subtracting the historical built-up area from the current built-up area, and then dividing the result by the historical built-up area, as shown in Figure 10."
The expansion range and expansion speed should be presented in formulas for quick understanding better.
7. Additionally, we examined the built-up area growth rate at an equal interval of 6 years from 1994 to 2018, which yielded results consistent with the original experimental results. However, to accurately reflect the actual situation of urban expansion in China, and using all corrected data, we analyzed 2000, 2005, and 2010 as nodes. These nodes can better capture the peak and trough periods of urban expansion and accurately reflect its speed and amplitude. The chart below shows the expansion range and speed at equal intervals of 6 years from 1994 to 2018.
The figure under the response looks quite different from Figure 10 although the period shifted only one year. This means the indicator is not useful because it requires careful manual tuning for capturing significant changes on ground.
I suggest presenting the indicators in equal intervals with your analysis and insights to the indicators, discussing advantages and limitations of the indicators.
Also, I suggest showing time-series charts of the built-up areas, which supports better understanding what changes the indicators represent.
Thank you in advance for your consideration.
